

# Fast-freezing with liquid nitrogen preserves bulk dissolved organic matter concentrations, but not its composition

Lisa Thieme[1,2], Daniel Graeber[3], Martin Kaupenjohann[1], Jan Siemens[2]

5  [1] Chair of Soil Science, Department of Ecology, Technical University of Berlin, Berlin,
[2] Chair of Soil Resources, Institute of Soil Science and Soil Conservation, iFZ Research Centre for Biosystems, Land Use and Nutrition, Justus-Liebig University Giessen, Giessen, Germany
[3] Department of Bioscience, Catchment Science and Environmental Management, Aarhus University, Silkeborg, Denmark

*Correspondence to*: L. Thieme (l.thieme@campus.tu-berlin.de)



**Abstract**. Freezing can affect concentrations and spectroscopic properties of dissolved organic matter (DOM) in water samples. Nevertheless, water samples are regularly frozen for sample preservation. In this study we tested the effect of different freezing methods (standard freezing at -18°C and fast-freezing with liquid nitrogen) on DOM concentrations

measured as organic carbon (DOC) concentrations and on spectroscopic properties of DOM from different terrestrial ecosystems (forest and grassland). Fresh and differently frozen throughfall, stemflow, litter leachate and soil solution samples were analyzed for DOC concentrations, UV-vis absorption and fluorescence excitation-emission matrices combined with parallel factor analysis (PARAFAC). Fast-freezing with liquid nitrogen prevented a significant decrease of DOC concentrations observed after freezing at -18°C. Nonetheless, the share of PARAFAC components 1 (EXmax <250 nm (340

nm), EMmax: 480 nm) and 2 (EXmax: 335 nm, EMmax: 408 nm) to total fluorescence and the humification index (HIX) decreased after both freezing treatments, while the shares of component 3 (EXmax: <250 nm (305 nm), EMmax: 438 nm) as well as $SUVA_{254}$ increased. The contribution of PARAFAC component 4 (EXmax: 280 nm, EMmax: 328 nm) to total fluorescence was not affected by freezing. We recommend fast-freezing with liquid nitrogen for preservation of bulk DOC concentrations of samples from terrestrial sources, whereas immediate measuring is preferable to preserve spectroscopic

properties of DOM**.**

**Keywords**

freezing, dissolved organic matter, fluorescence, absorption

## 1 Introduction

In addition to dissolved organic carbon (DOC) concentrations, properties of dissolved organic matter (DOM) are crucial for

its role in biogeochemical cycles of carbon and nutrients as well as for its effect on pollutant dynamics (Bolan et al., 2011). Spectroscopic methods like UV-vis absorption and fluorescence spectroscopy used as single excitation/emission scans, synchronous scans and excitation-emission matrices (EEMs) in combination with different indices and/or parallel factor analysis (PARAFAC) are increasingly applied to characterize chromophoric dissolved organic matter (cDOM) in various environments. Optical methods have been used to assess origin, dynamics, biogeochemical cycling and fate of cDOM in a

wide range of marine and freshwater systems (e.g., Murphy et al., 2008; Yamashita et al., 2010; Stedmon and Markager, 2005; Stedmon et al., 2007; Graeber et al., 2012). In addition to cDOM in samples from aqueous systems, water-extractable soil organic matter and cDOM in soil pore water samples (Otero et al., 2007; Hur et al., 2014; Traversa et al., 2014) were investigated using EEMs plus PARAFAC as well as isolated humic substances from soil and litter (Kalbitz et al., 1999; D'Orazio et al., 2014). The applicability of optical methods for characterizing DOM and the comparability of results in

multidisciplinary studies relies on the preservation of samples prior to their analysis. DOM properties depend on many



physico-chemical and biological boundary conditions, so that artefacts caused by sample storage or sample pre-treatment may easily be produced. For these reasons it is recommended to directly filter samples after collection and store them in the cold and dark prior to measurement as soon as possible (Spencer and Coble, 2014). However, immediate measurement is often not possible for practical reasons such as a large number of samples, remote or separated sampling sites, so that

freezing of filtered DOM samples is often the selected storage method (Murphy et al., 2008; Yamashita et al., 2010; Graeber et al., 2012). Studies on marine waters (Del Castillo and Coble, 2000; Yamashita et al., 2010; Conmy et al., 2009) showed only a small freezing effect on DOM fluorescence characteristics, but tests with a variety of freshwater samples produced significant changes in DOM composition, however with inconsistent results in terms of direction of the changes (e.g., Fellman et al., 2008; Yamashita et al., 2010; Spencer et al., 2007).

Apparently the impact of sample preservation like freezing is highly variable depending on sample and DOM characteristics. While most studies focused on samples from marine or freshwater ecosystems, there is a lack of information on sample pre-treatment effects on cDOM properties of water samples from terrestrial ecosystems. Here we investigate the influence of freezing and thawing on DOC concentration, spectral absorption and fluorescence properties for a wide range of water samples (throughfall, litter leachate and soil solution) from different terrestrial ecosystems (grasslands and forests). During

the freezing process, DOM is preferentially excluded from the ice phase and enriched in the remaining liquid phase (Belzile et al., 2002; Xue et al., 2015). The increasing solute concentrations and changing conditions like viscosity (Zaritzky, 2006) and pH (Shafique et al., 2011) in the remaining liquid phase during the freezing process could promote conformational and configurational changes of DOM molecules as well as particle and complex formation depending on DOM composition and sample type. We tested in how far fast-freezing with liquid nitrogen might prevent concentration and partitioning effects and

minimize the time for conformational changes of DOM. We hypothesized i) that sample type affects freeze/thaw effects on DOC concentrations and DOM properties, because of different physical and chemical DOM characteristics and therefore different response to changing conditions during freezing and ii) that fast-freezing with liquid nitrogen reduces these freeze/thaw effects, because it minimizes the freezing time and thus prevents partitioning effects and their physical consequences.

## 2 Material and methods

### 2.1 Study sites

The study was conducted on experimental plots in the Schorfheide Chorin Exploratory of the German "Biodiversity Exploratories", which were established as platform for large-scale and long-term functional biodiversity research (Fischer et al., 2010). The experimental plots are located in a young glacial landscape in NE Germany with an annual mean temperature

of 8 to  8.5°C and an annual mean precipitation of 500 to 600 mm. The forest plots are dominated either by pine (*Pinus sylvestris* L.) or beech (*Fagus sylvatica* L.) on Cambisols (IUSS working group WRB, 2014). The grassland plots are meadows, pastures and mown pastures on Histosols, Gleysols and Cambisols.



### 2.2 Sampling and sample preparation

For the experiments, five composite samples from grassland sites and 22 composite samples from forest sites were collected on 17 and 18 June 2014. Together we had 27 samples including six throughfall (TF), five stemflow (SF), five forest litter leachate (LL) as well as six top- and five subsoil solution samples. Volume-weighted composite samples were produced in

"aged" 500 ml PE bottles by merging samples of the same sample type per site. The bottles were biweekly used for the same samples on a routine sampling campaign of above- and below-ground water at the biodiversity exploratory sites, after washing in the dishwasher and with deionised water. TF was sampled with funnel-type collectors (diameter 0.12 m, polyethylene) 0.3 m above soil surface. We pooled five replicates at grassland and 20 replicates arranged in two lines of 10 samplers in a cross shaped form at forest sites. To minimize alterations of the sample and contamination such as evaporation,

photo chemical reactions and algae growth, the sampling bottles were wrapped with aluminium foil and closed with a 1.6 mm polyester mesh and a table-tennis ball. SF was sampled with sliced polyurethane hoses (diameter: 0.04 m) as a collar sealed with a polyurethane-based glue to the bark of three trees per site at approximately 1.5 m height and connected with a polypropylene (PP) or polyethylene (PE) barrel via a PE tube. LL was collected with three zero-tension lysimeters per site (280 cm$^2$ sampling area) consisting of polyvinyl chloride plates covered with a PE net (mesh width 0.5 mm) connected with

PE hoses to 2 L PE bottles stored in a box below ground. We sampled soil solution with nylon membrane (0.45 µm) suction cups (ecoTech, Germany). Three samplers were installed beneath the A horizon (Top) at approximately 10 cm depth. Another three were installed in the B horizon (Sub) in approximately 50 cm depth in the forest plots and 60 or 70 cm depth in the grassland sites. Suction cups were connected to 2 L PE bottles in an insulated aluminium box placed into a soil pit. Soil water was extracted by applying a vacuum of 50 kPa to the PE bottles with an electric pump after each sampling.

After mixing, the samples were transported on ice to the laboratory and stored overnight at 5°C. We measured pH (Knick, Germany) and electrical conductivity (WTW, Germany) in all samples prior to filtration through ~ 0.7 µm glass microfiber filters (Whatman GF/F). The filters were washed with 100 ml deionised water and 10 ml of sample before sample filtration. The filtered sample was split in three aliquots for different preservation treatments: i) no preservation (fresh) for which samples were stored at 5°C in the dark and DOC concentrations were measured 24 h after sampling while fluorescence as

well as absorbance were measured within 48 h; ii) preservation by freezing for which the samples were stored at -18°C for four weeks, and iii) fast-freezing with liquid nitrogen (N$_2$), for which 1 ml sample aliquots were filled in pre-rinsed 15 ml (5 ml sample) PP falcon tubes, dipped in liquid nitrogen for 30 s and then stored at -18 °C for 42 days. Fresh samples and samples frozen at -18°C were stored in 20 ml PE scintillation vials (NeoLab) that were pre-rinsed with 5 ml sample before filling. Fluorescence, absorbance and DOC concentration from all frozen samples were measured after defrosting over night

at 5 °C in the dark. For all preparation steps and treatments control samples of ultrapure water (EVOQUA, Germany) were analyzed.





### 2.3 Laboratory analysis

We measured the concentration of DOC as non-purgeable organic carbon on a Shimadzu TOC-5050A (Duisburg, Germany) with a limit of quantification of 2 mg $L^{-1}$. Absorption spectra of DOM were scanned at wavelength from 400 to 600 nm using a Lambda 20 UV-vis spectrometer (Perkin Elmer, USA) and a 1 cm quartz cuvette. Absorbance measurements were

baseline corrected using ultrapure water. All fluorescence EEMs were measured on a Hitachi F-4500 fluorescence spectrometer (Hitachi, Japan) directly after absorption measurement in the same cuvette. We measured excitation from 240 to 450 nm (5 nm steps) and emission from 300 to 600 nm (2 nm steps) with a slit width of 5 nm and scan speed 12000 nm $min^{-1}$. We corrected our EEMs according to the protocol from Murphy (2010) with the fdomcorrect function in the drEEM toolbox (version 2.0) of Murphy et al. (2013) using Matlab (Version Matlab2011b, The MathWorks Inc.). We used the

supplies provided by the manufacturer for the excitation and emission correction factors. We measured ultrapure water fluorescence spectra for blank correction and to convert EEMs to Raman units by normalizing them to the area under the Raman peak at 350 nm excitation wavelength (Lawaetz and Stedmon, 2009). In order to apply the inner-filter correction of Lakowicz (2006) integrated in the drEEM toolbox, all aliquots were diluted with ultrapure water to achieve an absorption of <0.3 at 254 nm (Ohno, 2002). For this reason, not all treatments of one sample were diluted with the same dilution factor.

To test the possible influence of different dilutions on the pH-related changes in fluorescence (Patel-Sorrentino et al., 2002; Baker et al., 2007), dilution series with samples (n = 14) from the same plots and same sample types but with different sampling dates where measured for pH, absorption and fluorescence according to the protocol described above. We compared the differences of 31 dilutions and calculated the mean absolute deviation (MAD). These were compared to the MAD of measurement precision, determined by analysing 11 samples in three replications. For the PARAFAC components

%C1, %C2 and %C3 and $SUVA_{254}$ the MAD caused by dilution were less or equal than the precision MAD, so that there was no influence of dilution on the three humic-like components and the specific UV absorbance at 254 nm. For %C4 and HIX the effect of dilution could exceed the precision of fluorescence measurements. For detailed information see supporting information.

**2.4 Spectroscopic indices and PARAFAC modelling**

Based on the absorbance spectra, we calculated specific ultraviolet absorbance ($SUVA_{254}$) as the absorbance at 254 nm divided by the DOC concentration. The $SUVA_{254}$ is reported in L $mg^{-1}$ $m^{-1}$, and is associated with bulk aromaticity (Weishaar et al., 2003). Moreover, we calculated the humification index (HIX) from fluorescence EEMs (Ohno, 2002). The HIX ranges from 0 to 1 and allows characterizing samples based on their degree of DOM humification.

In addition to the calculation of indices, we used parallel factor analysis (PARAFAC) to mathematically decompose the trilinear data of the EEMs into fluorescence components of DOM (Stedmon et al., 2003). Further pre-processing steps of EEMs (smoothing of Rayleigh and Raman scatter and sample normalization), as well as the PARAFAC analysis were





conducted with the drEEM toolbox (version 2.0, Murphy et al., 2013). We chose a four component PARAFAC model (components referred as C1-C4), visually checked the randomness of residuals and the component spectral loadings, split-half validated the model and generated the best fit by random initialization. For comparison in statistical analysis we used the relative percentage distribution of the four PARAFAC components (% of the sum of total peak fluorescence of all

PARAFAC components), so that percentage values for the components will be given as %C1 to %C4.

### 2.5 Statistical analysis

The DOM composition variables used for statistical analysis were the PARAFAC components %C1 to %C4, the spectroscopic indices HIX and $SUVA_{254}$, as well as the DOC concentration. For all statistical analysis the variables were scaled and centred. We conducted a pair-wise (samples as strata) permutational multivariate analysis of variance

(PERMANOVA) based on Euclidean distances in R (Oksanen et al., 2015; R core team, 2015). The adonis function was used to assess the influence of sample preparation (fresh, frozen, fast-freezing) on DOM variables. To investigate preservation effects on single variables we conducted linear mixed-effect models (sometimes called multi-level models, lme function, Linear and Nonlinear Mixed Effects Models package for R, Pinheiro et al., 2015) with samples as random intercept on each of the DOM composition variables. These were used instead of simple linear models or ANOVAs, since we could

not expect the same intercept for all samples due to different sample concentrations. To assess the influence of sample type (TF, SF, LL, Top or Sub) on the relative change of DOM composition due to fast-freezing with liquid nitrogen or freezing at -18°C in relation to the measurement of fresh, cooled samples, we used an ANOVA with the sample type as fixed factor (aov function in R). To remove sample concentration-related effects and to calculate relative changes, the differences between the two preservations (either fast-freezing or freezing at -18°C) relative to the measurements of fresh samples were calculated

for each sample before the ANOVA. This was only done for variables, for which we found strong, significant effects with the linear mixed-effect models.

### 3 Results

#### 3.1 DOM concentrations

The samples covered a wide range of DOC concentrations (Fig. 1a, b). Fresh TF samples showed the lowest concentrations

ranging from 5 to 17 mg $L^{-1}$, SF samples had the highest DOC concentrations ranging from 12 to 138 mg $L^{-1}$ (Fig. 1b). High concentrations up to 75 mg $L^{-1}$ were also found for LL samples, but average values were smaller than for SF (Fig. 1b). In the mineral soil, concentrations decreased from 13 to 124 mg $L^{-1}$ in topsoil samples to 9 to 47 mg $L^{-1}$ in subsoil samples.

We found a significant treatment effect (linear mixed-effect models (lme), $p<0.05$) on DOC concentration when comparing the fresh and frozen samples (Fig. 1c). In 24 of 27 samples DOC concentrations decreased after freezing at -18°C and

subsequent thawing, with an average change of - 1.6 mg $L^{-1}$ or - 6% respectively. The maximum decrease that was found





equalled - 6 mg L$^{-1}$ and - 25%, respectively. In contrast to freezing at -18°C, fast-freezing with liquid nitrogen did not result in significant changes (lme, p>0.05) of DOC concentrations (Fig. 1c).

### 3.2 PARAFAC fluorescence components

The analysis of fluorescence spectra using PARAFAC resulted in four components that were characterized according to the

review of Fellman et al., (2010) (Table 1). C1 exhibited its main excitation maximum at < 250 nm, a secondary maximum at 340 nm and an emission maximum at 480 nm and was described as UVA humic-like fluorophore with a terrestrial source and a high molecular weight (Murphy et al., 2006; Stedmon et al., 2003; Shutova et al., 2014; Fellman et al., 2010). C2 had a maximum excitation at 335 nm and an emission maximum at 408 nm and was named also UVA humic-like, but associated with low molecular weight (Murphy et al., 2006; Fellman et al., 2010; Stedmon et al., 2003). C3 was defined by an

excitation maximum at < 250 nm, a secondary maximum at 305 nm and an emission maximum at 438 nm. This component dominated fulvic acid fractions of humic substances (Santín et al., 2009; He et al., 2006). Finally, C4 was characterized by its excitation maximum at 280 nm and an emission maximum at 328 nm and was classified as tryptophan-like, as its fluorescence resembles free tryptophan. Therefore, this component was associated with free or bound proteins (Fellman et al., 2010).

We found different distributions of PARAFAC components for different sample types (Fig. 2). The contribution of %C1 to the total fluorescence increased from TF over SF to LL and then decreased again from LL to Sub (Fig. 2), while %C2 showed just the opposite trend. In contrast, %C3 tended to increase from TF to Sub, whereas %C4 showed a decreasing trend (Fig. 2).

Similar changes in component distribution were found as a consequence of freezing at -18°C and fast-freezing with liquid

nitrogen (Fig. 3). We observed a significant (lme, p<0.05) decrease in all samples for the relative fraction of the humic-like components %C1 and %C2 after freezing at -18°C and fast-freezing compared to the fresh control samples (Fig. 3a, b). The contribution of %C1 to the total fluorescence decreased on average by -3% with maximum changes of -5% for freezing at -18°C and -6% for fast-freezing with liquid nitrogen. The average decrease of %C2 was -3% and the maximum -8% for both treatments.

In contrast to %C1 and %C2, the share of %C3 to the total fluorescence intensity increased upon freezing (Fig. 3e, f). All samples frozen at -18°C showed an increase in the relative intensity of the %C3 signal, with an average increase of +6% for both treatments. The maximum increase was +10% (-18°C) and +12% (N$_2$). No significant effects of sample preservation (lme, p>0.05) were found for %C4, the protein-like-component (Fig. 3g, h).

### 3.3 Aromaticity and humification index

We found SUVA$_{254}$-values ranging from 1.1 L mg$^{-1}$ m$^{-1}$ up to 4.5 L mg$^{-1}$ m$^{-1}$ for fresh samples (Fig. 4a, b). Samples frozen at -18°C and fast-frozen samples showed a significant increase (lme, p<0.05) of their SUVA$_{254}$ (Fig. 4c). The average change





was +0.4 L mg$^{-1}$ m$^{-1}$ equivalent to +20% for samples frozen at -18°C and +0.5 L mg$^{-1}$ m$^{-1}$ equivalent to +24% for samples that were fast-frozen with liquid nitrogen.

The humification index of the freshly measured samples ranged from 0.806 to 0.931 in TF and SF samples and from 0.849 to 0.975 for Sub, Top and LL samples (Fig. 5a, b). We found a significant decrease (lme, p<0.05) of the HIX when comparing the freshly measured samples with the frozen and the fast-frozen samples (Fig. 5c). The average change was -0.016 or -2% for samples frozen at -18°C and -0.020 or -2% for samples fast-frozen with liquid nitrogen. The maximum decrease was -0.128 or -15% for -18°C samples and -0.076 or -8% for liquid nitrogen samples (Fig. 5 c, d, e, f).

## 4 Discussion

We found that freezing at -18°C significantly reduced DOC concentrations across all sample types. This is in line with results of Fellman et al. (2008) investigating the effect of freezing and thawing on Alaskan stream water samples. This loss of DOC concentration might be due to aggregation and irreversible particle formation (Giesy and Briese, 1978) induced by partitioning and concentration effects during the freezing process (Belzile et al., 2002; Xue et al., 2015). Indeed, our results indicated that fast-freezing with liquid nitrogen can prevent significant reductions of DOC concentrations, mainly because topsoil and subsoil solution DOC concentrations were less affected than during freezing at -18°C.

In contrast to effects on DOC concentrations, we found similar significant effects of fast-freezing as well as freezing at -18°C on the chromophoric humic fraction of DOM (PARAFAC components, HIX and SUVA$_{254}$). The increase of aromaticity as indicated by higher SUVA$_{254}$ values indicates a stronger removal of non-aromatic DOM during freezing and thawing. On the other hand, the decrease in the HIX suggests a preferential removal of humified cDOM. One potential explanation for the fact that fast-freezing in liquid nitrogen resulted in significant changes of DOM fluorescence properties, but only small changes of bulk DOC concentrations, is that cDOM reacted stronger to freezing and thawing than the remaining DOM so that spectroscopic properties were affected, but bulk DOC concentrations were not. Fast freezing may have failed to prevent changes of cDOM composition because i) cDOM changes occurred not only during the freezing process (-18°C or -196°C in liquid nitrogen), but also in frozen state at -18°C in the freezer during storage or ii) cDOM was affected by the thawing process that was identical for both freezing treatments. The formerr might be supported by a recrystallisation of ice crystals in frozen state (Luyet, 1967; Meryman, 2007).

No significant changes of protein-like fluorescence (%C4) due to freezing and thawing were observed. This is in contrast to the results of Spencer et al. (2007), which could be related to similar fluorescence characteristics, but different chemical composition of proteinaceous fluorescence material from aquatic sources and the solutions from terrestrial ecosystems tested in this study.



## 5 Conclusion

Freezing and thawing affected the DOC concentration, spectral absorption and fluorescence properties of water samples (throughfall, litter leachate and soil solution) from different terrestrial ecosystems (grasslands and forests). In contrast fast-freezing with liquid nitrogen minimized the changes of bulk DOC concentrations but not the changes of spectroscopic

cDOM properties. Different thawing protocols for minimizing sample storage effects on DOM should be tested in future studies. We suggest the use of fast-freezing for preservation of bulk DOC concentrations of samples, but normal freezing or fast-freezing should be avoided to preserve cDOM characteristics of samples from terrestrial sources. Instead, filtration, cooling and measurements soon after the sampling should be the method of choice, if possible.

**Data availability**

The data is available in the supplementary information

**Author contribution**

L.Th, M.K., and J.S. designed the experiment, L.T.h performed the experiments. All authors analyzed the data and wrote the manuscript.

**Acknowledgements**

We thank the managers of the three Exploratories, Kirsten Reichel-Jung, Swen Renner, Katrin Hartwich, Sonja Gockel, Kerstin Wiesner, and Martin Gorke for their work in maintaining the plot and project infrastructure; Christiane Fischer and Simone Pfeiffer for giving support through the central office, Michael Owonibi for managing the central data base, and Markus Fischer, Eduard Linsenmair, Dominik Hessenmöller, Jens Nieschulze, Daniel Prati, Ingo Schöning, François Buscot, Ernst-Detlef Schulze, Wolfgang W. Weisser and the late Elisabeth Kalko for their role in setting up the Biodiversity

Exploratories project.

The work has been (partly) funded by the DFG Priority Program 1374 "Infrastructure-Biodiversity-Exploratories" (SI 1106/4-1,2). D. Graeber was supported by a grant from the Danish Centre for Environment and Energy, Aarhus University. Field work permits were issued by the responsible state environmental offices of Baden-Württemberg, Thüringen, and Brandenburg (according to § 72 BbgNatSchG). We thank Sabine Dumke and Robert Jonov for sample measurement.



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



**Tables**

**Table 1: Characteristics of PARAFAC components based on Fellman et al., 2010**

| Component | Maximum exitation wavelength($EX_{max}$) (nm) | Maximum emission wavelength ($EM_{max}$) (nm) | Description |
|---|---|---|---|
| C1 | <250 (340) | 480 | humic-like, terrestrial |
| C2 | 335 | 408 | humic-like |
| C3 | <250 (305) | 438 | fulvic-acid-type |
| C4 | 280 | 328 | tryptophan-like |





**Figure captions**

Figure 1: Absolute DOC concentrations (measured in fresh samples) and changes of DOC concentrations after freezing (-18°C) and fast-freezing with liquid nitrogen; a, c, e: all samples (n= 27); b, d, f: ordered by sample type (throughfall (TF) n=6, stemflow (SF) n=5, litter leachate (LL) n=5, top soilsolution (Top) n=6, sub-soilsolution (Sub) n=5); gray dashed line:

analytical reproducibility; *** significant changes (linear mixed models (lme), p<0.05); Boxplots: solid line: median, dashed line: mean

Figure 2: Mean distribution of PARAFAC components %C1-%C4 for different sample types

Figure 3: Changes of relative distribution of PARAFAC components after freezing (-18°C) and fast-freezing with liquid nitrogen; a, c, e, g: all samples (n=27); b, d, f, h ordered by sample type (throughfall (TF) n=6, stemflow (SF) n=5, litter leachate (LL) n=5, top soilsolution (Top) n=6, sub-soilsolution (Sub) n=5); gray dashed line: analytical reproducibility; *** significant changes (linear mixed models (lme), p<0.05) ;Boxplots: solid line: median, dashed line: mean

Figure 4: Absolute values (measured in fresh samples) and changes of SUVA254 after freezing (-18°C) and fast-freezing with liquid nitrogen; a, c, e: all samples (n= 27); b, d, f: ordered by sample type (throughfall (TF) n=6, stemflow (SF) n=5, litter leachate (LL) n=5, top soilsolution (Top) n=6, sub-soilsolution (Sub) n=5); gray dashed line: analytical reproducibility; *** significant changes (linear mixed models (lme), p<0.05); Boxplots: solid line: median, dashed line: mean

Figure 5: Absolute values (measured in fresh samples) and changes of HIX after freezing (-18°C) and fast-freezing with liquid nitrogen; a, c, e: all samples (n= 27); b, d, f: ordered by sample type (throughfall (TF) n=6, stemflow (SF) n=5, litter leachate (LL) n=5, top soilsolution (Top) n=6, sub-soilsolution (Sub) n=5); gray dashed line: analytical reproducibility; *** significant changes (linear mixed models (lme), p<0.05); Boxplots: solid line: median, dashed line: mean

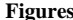

**Figures**

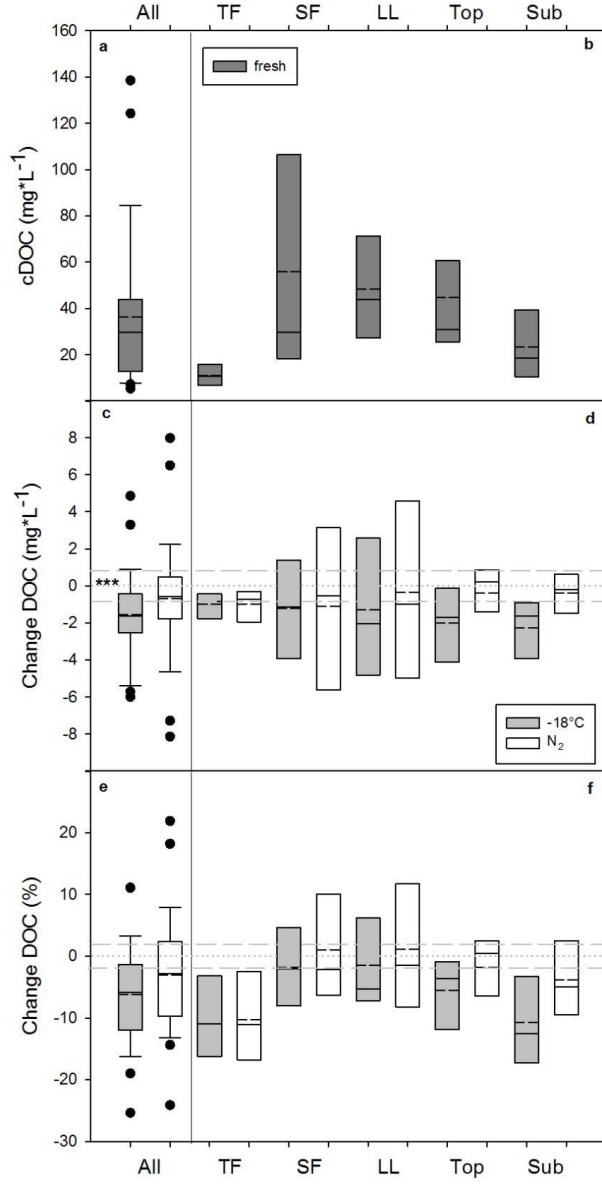

Figure 1




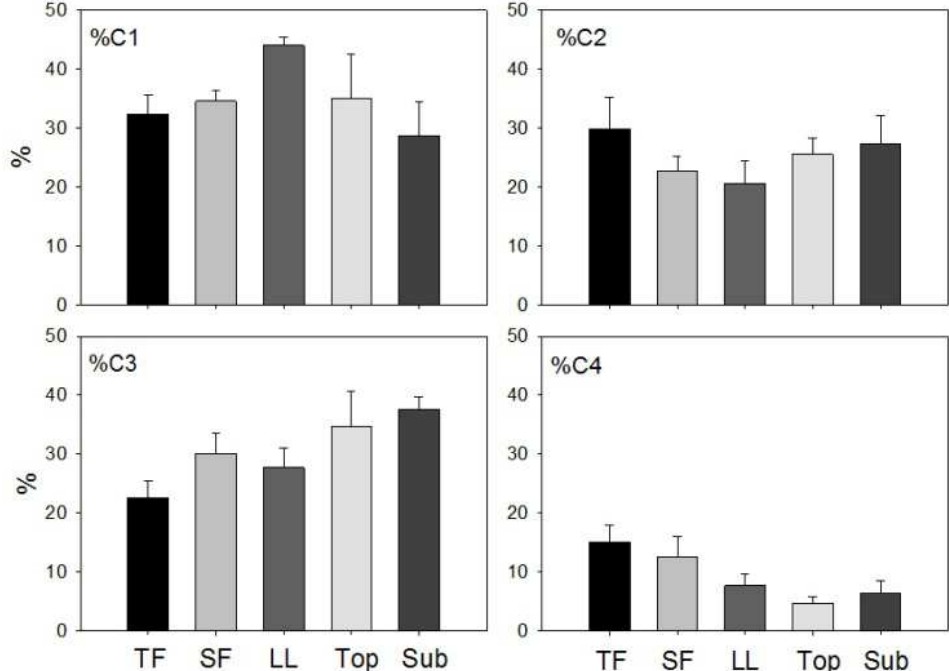

**Figure 2**



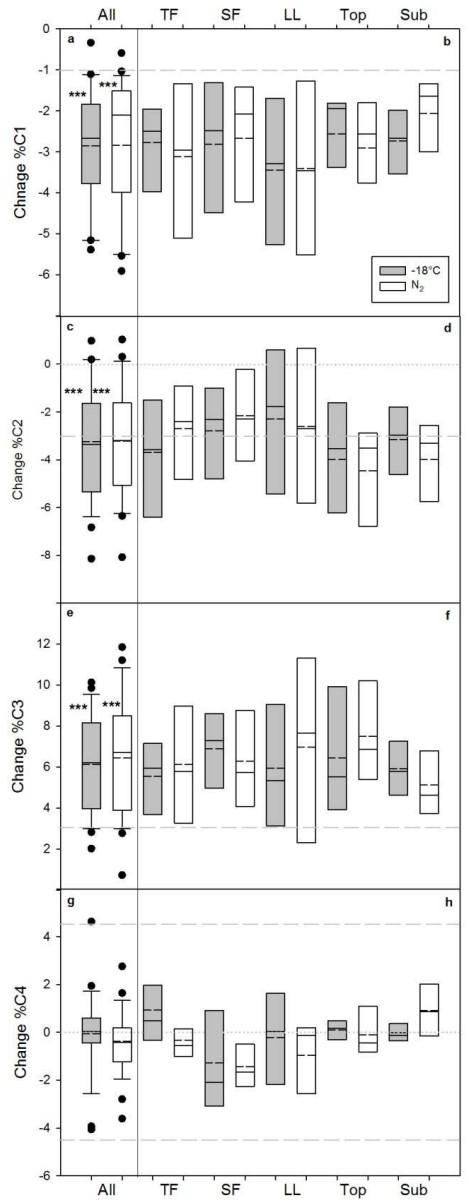

**Figure 3**





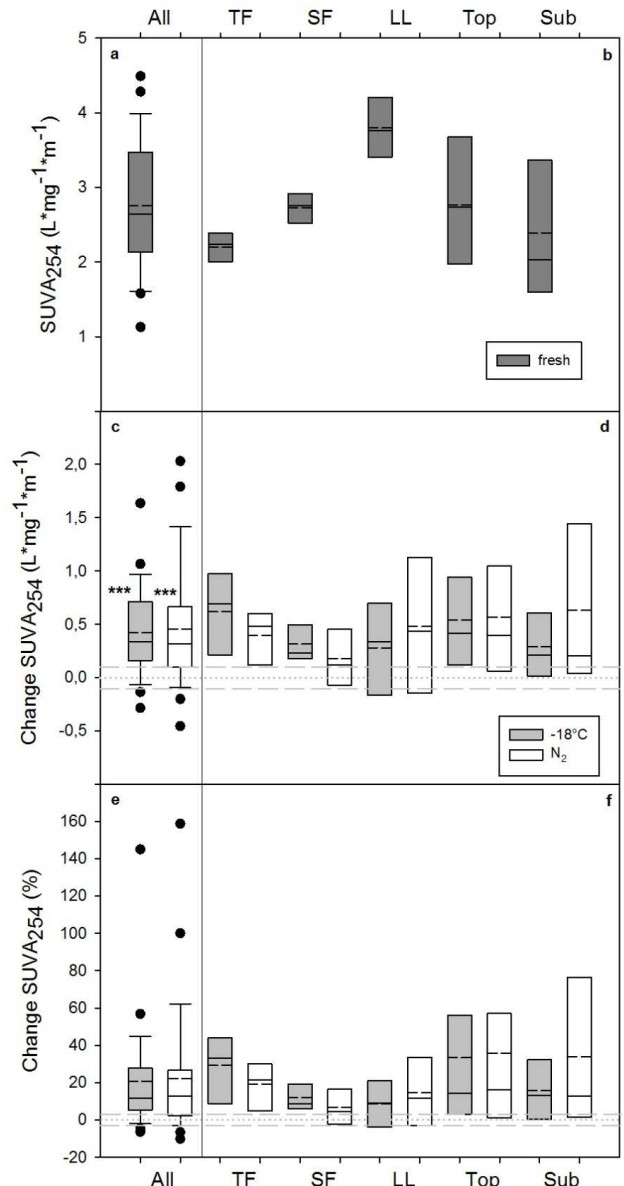

**Figure 4**





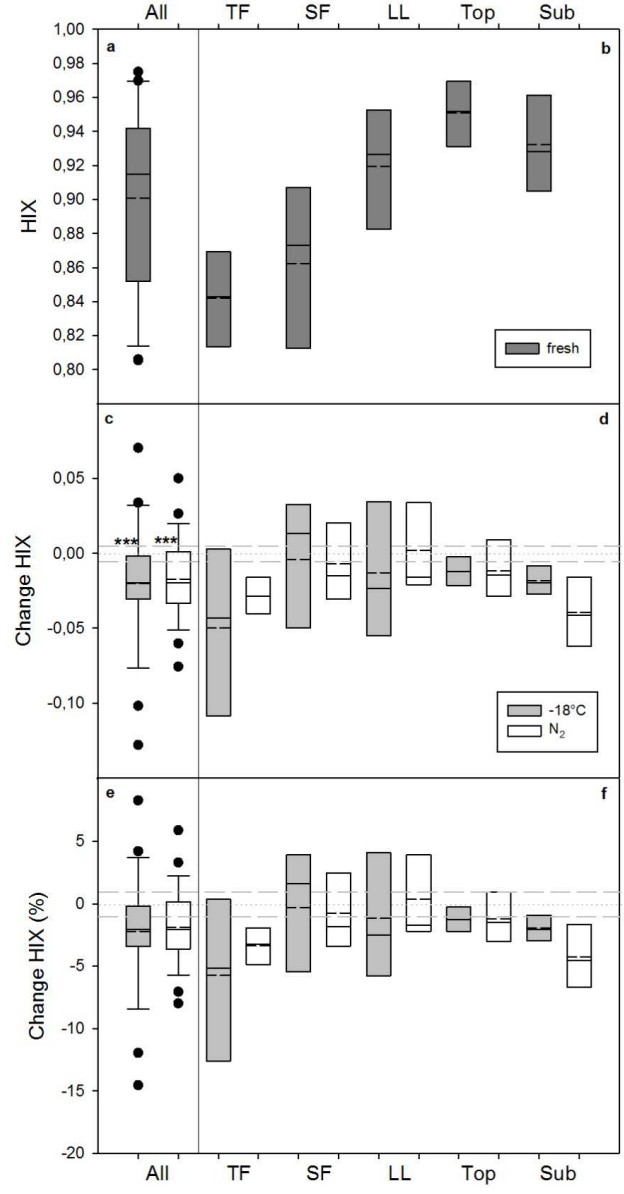

**Figure 5**