# Peer review of "Fast-freezing with liquid nitrogen preserves bulk dissolved organic matter concentrations, but not its composition"

_Biogeosciences, 2016_

## Referee Comment (RC1) · Anonymous Referee #1 · 30 Mar 2016

This paper has dealt with effects of freezing types on DOM concentrations and properties. The manuscript is well-written and organized. The topic is useful for reseachers related to DOM dynamics in terrestrial and aquatic ecosystems.

It is well-known that freezing can decrease DOC concentration due to aggregation and irreversible particle formation, but this paper reported that liquid nitrogen can reduce concentration changes. This will be useful for the readers. On the other hand, composition changes were detected even for samples freezed by liquid nitrogen.Freezing can cause selective removal of non-aromatic DOM during freezing and thawing. This will be inevitable using freezing of samples.

In the present version of manuscript, several mistakes need to be revised.

[Figure]

P4L5ãĂĂRevise "500 ml" to "500 mL". P4L26, 28ãĂĂRevise "ml" to "mL". P6L25 Revise "mg L-1" to "mg C L-1". P7L5 Revise "Fellman et al., (2010)" to "Fellman et al. (2010)" P8L24 Revise "formerr" to "former"

---

## Referee Comment (RC2) · Anonymous Referee #2 · 14 Apr 2016

The paper focus on evaluating the effects of fast-freezing with liquid nitrogen and of freezing at -18°C on DOC and DOM contents of water samples from different terrestrial ecosystems. In my opinion the manuscript contains important and useful results for publication. However, the manuscript presents some minor points that should be addressed in a revised version, which are following presented.

1. In section "Abstract", the last sentence highlight important findings "We recommend fast-freezing with liquid nitrogen for preservation of bulk DOC concentrations of samples from terrestrial sources, whereas immediate measuring is preferable to preserve spectroscopic properties of DOM." However, the last part of the sentence was also suggested by the study of Santos et al. (2010) for bulk deposition samples (rainwater

samples), which show that such study should be used in the discussion of the present manuscript.

2. In section "1 Introduction", page 2, reformulate the sentence "In addition to cDOM in samples from aqueous systems, water-extractable soil organic matter and cDOM in soil pore water samples (Otero et al., 2007; Hur et al., 2014; Traversa et al., 2014) were investigated using EEMs plus PARAFAC as well as isolated humic substances from soil and litter (Kalbitz et al., 1999; D'Orazio et al., 2014)." The study of Otero et al. (2007) did not used the EEMs plus PARAFAC as well as isolated humic substances from soil and litter.

3. In section "1 Introduction", page 3, I suggest to add also the reference of Santos et al. (2010) to the following sentence "For these reasons it is recommended to directly filter samples after collection and store them in the cold and dark prior to measurement as soon as possible (Spencer and Coble, 2014)".

4. In section "2 Material and methods", subsection "2.2 Sampling and sample preparation", page 4, the first and fourth sentences seems to be contradictory, because it is presented that samples were collected on 17 and 18 June 2014, and then is presented that bottles were biweekly used. Please, clarify.

5. In section "2 Material and methods", subsection "2.2 Sampling and sample preparation", page 4: why were not used glass bottles and vials to store the samples? Glass material should be used to avoid contaminations. Blanks of procedure were performed?

6. In section "3 Results", subsection "3.2 PARAFAC fluorescence components", reformulate the sentence "The maximum increase was +10% (-18°C) and +12% (N2)". Remove the plus sign and extend the sentence with the types of freeze.

7. In section "4 Discussion", the reference of Santos et al. (2010) should be used together with the reference to Spencer et al. (2007) to the following sentence "This

is in contrast to the results of Spencer et al. (2007), which could be related to similar fluorescence characteristics, but different chemical composition of proteinaceous fluorescence material from aquatic sources and the solutions from terrestrial ecosystems tested in this study."

---

## Referee Comment (RC3) · Anonymous Referee #3 · 17 May 2016

Freezing dissolved organic matter was a common technique and recently has come under increased scrutiny due to potential impacts to the solution chemistry and chemical character of compounds in solution. There is a growing body of literature on the influence of freezing on streamwater, but apparently not any experiments to my knowledge that treat this problem in soil solution. The paper by Thieme et al. appears to be the first attempt to investigate the issue of freezing soil solution and therefore, is a compelling topic that could benefit the research community working in soil and stream dissolved organic matter. Overall, the manuscript has a solid foundation, but there are several areas where the authors could improve the manuscript. I feel there is also a major shortcoming that may not be egregious enough to prevent publication, but

represents a major confounding issue in the interpretation of the results. The major shortcoming in my view is the method of compositing both grassland and forest soils together for analysis. These soil systems are very different and one would expect them to behave quite differently in terms of the dissolved organic matter produced within the soil matrix as well as exported from the site. Investigating the results of the freezing on these samples separately is a missed opportunity and I suggest that it should be a priority to do a follow-up to see if the results are similar or some important patterns in the results have been masked by the increased level of variability due to the composite samples of these two soil types. General comments: Introduction: The introduction focuses a great deal on fluorescence, while not mentioning the experimental approach of freezing until much later in the discussion. The extensive literature review on fluorescence isn't necessary given the common nature of the technique and the focus of the paper. I suggest reducing the discussion of fluorescence and spending more time summarizing current research on freezing and identifying knowledge gaps in this area. Specifically, I think it would be important to see if any experiments have been conducted on freezing soil solution. Highlighting the novelty of the approach is critical for the impact of this study. In addition, keep the information of freezing organic matter in general. Some time could also be dedicated to discussing what might be different between stream samples and soil samples after freezing. Finally, a clear justification and rationale for the study needs to be part of the introduction. Sampling and sample preparation: The approach for sampling, replication, and defining the subject for the analysis needs to be clarified. The existing description is hard to follow. It might help to provide a diagram for where the samples originate and their fate, with a clear identification of what is composited and analyzed. This will clearly highlight the mixing of the grassland and forest samples. The freezing procedures are somewhat tedious. Is this operational? What happens if a large quantity of water is stored? Is there a potential difference given the small amounts used as test subjects in the study? Results: The overall average change of 6% (1.6 mg L-1) seems small given the high DOC concentrations in the samples. Is the lower average a result of the composite? L30: This doesn't

make sense. SUVA values increase, so aromatic compounds or aromaticity increase. But, humification index decreases? Conclusion: There needs to be some discussion of the results related to very high DOC concentrations in the sample. What are the implications for changes in the DOM character with freezing? Also, is freezing with N2 practical? Figure 1: Is cDOC an accepted convention? A label of DOC with the units usually implies a concentration. I suggest adding 'in' for Change in DOC concentration, Or DOC change.

---

## Author Comment (AC1) · 15 Jun 2016

We thank reviewer #1 for the constructive hints. Please find below our response.

In the revised manuscript we will correct the mentioned mistakes. We will change "ml" to "mL" when necessary, "mgL-1" to "mg C L-1" and "former" to "former". We will also correct the wrong comma setting in Fellman et al., (2010) to Fellman et al. (2010).

---

## Author Comment (AC2) · 15 Jun 2016

We thank referee #2 for the constructive comments. Please find below our response

1. In section "Abstract", the last sentence highlight important findings "We recommend fast-freezing with liquid nitrogen for preservation of bulk DOC concentrations of samples from terrestrial sources, whereas immediate measuring is preferable to preserve spectroscopic properties of DOM." However, the last part of the sentence was also suggested by the study of Santos et al. (2010) for bulk deposition samples (rainwater samples), which show that such study should be used in the discussion of the present manuscript.

We will consider and cite Santos et al (2010) in the revised version of our manuscript (see point 3 and 7).

2. In section "1 Introduction", page 2, reformulate the sentence "In addition to cDOM in samples from aqueous systems, water-extractable soil organic matter and cDOM in soil pore water samples (Otero et al., 2007; Hur et al., 2014; Traversa et al., 2014) were investigated using EEMs plus PARAFAC as well as isolated humic substances from soil and litter (Kalbitz et al., 1999; D'Orazio et al., 2014)." The study of Otero et al. (2007) did not used the EEMs plus PARAFAC as well as isolated humic substances from soil and litter.

Thank you for the hint. We will reorganized the introduction and rephrase the sentence into: "Spectroscopic methods like UV-vis absorption and fluorescence spectroscopy used as single excitation/emission scans, synchronous scans and excitation-emission matrices (EEMs) in combination with different indices and/or parallel factor analysis (PARAFAC) are increasingly applied to characterize chromophoric dissolved organic matter (cDOM) in various environments (e.g., Murphy et al., 2008; Yamashita et al., 2010; Stedmon and Markager, 2005; Graeber et al., 2012; Otero et al., 2007, Traversa et al., 2014, Kalbitz et al., 1999)."

3. In section "1 Introduction", page 3, I suggest to add also the reference of Santos et al. (2010) to the following sentence "For these reasons it is recommended to directly filter samples after collection and store them in the cold and dark prior to measurement as soon as possible (Spencer and Coble, 2014)".

We strongly agree with the reviewer on the importance of immediate filtration, as well as cold and dark storage. In our experiment samples were immediately filtered and stored cold in the dark. The reference Santos et al. (2010) will be added in the respective sentence in front of "Spencer and Coble, 2014".

4. In section "2 Material and methods", subsection "2.2 Sampling and sample preparation", page 4, the first and fourth sentences seems to be contradictory, because it is

presented that samples were collected on 17 and 18 June 2014, and then is presented that bottles were biweekly used. Please, clarify.

The samples for the cDOM storage experiment described in this manuscript were taken within a biweekly sampling routine of above and belowground water samples. It takes two days to collect samples from all research sites. Therefore we state in the Materials and Methods section that samples were taken on the 17th and 18th of June. For the in-field sample collection we use the same PE bottles for the same sample every 14 days. We will rephrase the respective paragraph into: "For the experiments, we collected solution samples from five forest and three grassland plots on 17 and 18 June 2014 within a bi-weekly 2 day sampling routine of above and below-ground water samples in the DFG priority programm "Biodiversity Exploratories". Together we collected 27 samples for the freezing experiment including six throughfall (TF), five stemflow (SF), five forest litter leachate (LL) as well as six top- and five subsoil solution samples. Volume-weighted composite samples were produced from replicated samplers of the same type (e.g. throughfall collectors, shallow suction cups) of one plot for the experiment in "aged" 500  ml PE bottles. The bottles were bi-weekly used in the field for the same samples, after washing in the dishwasher and with deionised water."

5. In section "2 Material and methods", subsection "2.2 Sampling and sample preparation", page 4: why were not used glass bottles and vials to store the samples? Glass material should be used to avoid contaminations. Blanks of procedure were performed?

Since we froze the samples, glass bottles could not be used because they could break when the water sample expands during the freezing process. For collecting the samples, we used aged HDPE bottles, which do not release detectable amounts of DOM according to our experience. We had blanks for all steps of the experiment. For all of them, no detectable DOC release (concentrations) and fluorescence was detectable. We will add this information to the Materials and Methods section of the revised manuscript and the data in table form in the supporting information.

6. In section "3 Results", subsection "3.2 PARAFAC fluorescence components", reformulate the sentence "The maximum increase was +10% (-18âŮȩC) and +12% (N2)". Remove the plus sign and extend the sentence with the types of freeze.

Thank you for the indication, we will rephrase the sentence accordingly.

7. In section "4 Discussion", the reference of Santos et al. (2010) should be used together with the reference to Spencer et al. (2007) to the following sentence "This is in contrast to the results of Spencer et al. (2007), which could be related to similar fluorescence characteristics, but different chemical composition of proteinaceous fluorescence material from aquatic sources and the solutions from terrestrial ecosystems tested in this study."

The reference Santos et al. (2010) will be added. We will rephrase the sentence into: "This is in contrast to the results of Spencer et al. (2007) and Santos et al. (2010), which could be related to similar fluorescence characteristics, but different chemical composition of proteinaceous fluorescence material from aquatic sources, rainwater and the solutions from terrestrial ecosystems tested in this study."

---

## Author Comment (AC3) · 15 Jun 2016

We thank referee #3 for the constructive comments. Please find below our response.

Introduction: The introduction focuses a great deal on fluorescence, while not mentioning the experimental approach of freezing until much later in the discussion. The extensive literature review on fluorescence isn't necessary given the common nature of the technique and the focus of the paper. I suggest reducing the discussion of fluorescence and spending more time summarizing current research on freezing and identifying knowledge gaps in this area. Specifically, I think it would be important to see if any experiments have been conducted on freezing soil solution. Highlighting the novelty of the approach is critical for the impact of this study. In addition, keep the

[Figure]

information of freezing organic matter in general. Some time could also be dedicated to discussing what might be different between stream samples and soil samples after freezing. Finally, a clear justification and rationale for the study needs to be part of the introduction.

We will reduce the discussion of fluorescence and review the scientific literature on the effect of freezing on fluorescence characteristics in water samples. We will include a clearer justification and rationale for the study. We will rephrase the introduction as follows: "In addition to dissolved organic carbon (DOC) concentrations, properties of dissolved organic matter (DOM) are crucial for its role in biogeochemical cycles of carbon and nutrients as well as for its effect on pollutant dynamics (Bolan et al., 2011). Spectroscopic methods like UV-vis absorption and fluorescence spectroscopy used as single excitation/emission scans, synchronous scans and excitation-emission matrices (EEMs) in combination with different indices and/or parallel factor analysis (PARAFAC) are increasingly applied to characterize chromophoric dissolved organic matter (cDOM) in various environments (e.g., Murphy et al., 2008; Yamashita et al., 2010; Stedmon and Markager, 2005; Graeber et al., 2012; Otero et al., 2007, Traversa et al., 2014, Kalbitz et al., 1999). The applicability of optical methods for characterizing DOM and the comparability of results in multidisciplinary studies relies on the preservation of samples prior to their analysis. DOM properties depend on many physicochemical and biological boundary conditions, so that artefacts caused by sample storage or sample pre-treatment may be produced easily. For these reasons it is recommended to directly filter samples after collection and store them in the cold and dark prior to measurement as soon as possible (Santos et al. 2010; Spencer and Coble, 2014;). However, immediate measurement is often not possible for practical reasons such as a large number of samples, remote or separated sampling sites, so that freezing of filtered DOM samples is often the selected storage method (Murphy et al., 2008; Yamashita et al., 2010; Graeber et al., 2012). Freezing can affect the physicochemical composition of samples (Edward and Cresser, 1992) so that improved conservation techniques, which avoid or minimize potential artifacts of freezing, are required. During

the freezing process, DOM is preferentially excluded from the ice phase and enriched in the remaining liquid phase (Belzile et al., 2002; Xue et al., 2015). The increasing solute concentrations and changing physical conditions in the remaining liquid phase during the freezing process could promote conformational and configurational changes of DOM molecules as well as particle and complex formation depending on DOM composition and sample type (Zaritzky, 2006; Edward and Cresser 1992). One potential technique for minimizing these effects could be fast freezing with liquid N2, by radically reducing the freezing time. Whereas studies on sample preservation of marine waters (Del Castillo and Coble 2000, Yamashita et al. 2010a, Conmy et al. 2009) showed only a small freezing effects on DOM fluorescence characteristics, research with a variety of freshwater samples produced inconsistent results. Fellman et al. 2008 measured DOC concentrations and UV absorption in fresh and frozen/thawed Alaska stream water samples and reported a significant decrease of DOC concentration and specific ultraviolet absorption at 254nm (SUVA254). They recommended freezing as an acceptable storage method for freshwater samples with low DOC concentration and/or low SUVA254 values. In contrast, Yamashita et al. 2010 observed only minor changes in absorption based indices after freezing and thawing of Venezuela river water but significant alterations (decrease and increase) for PARAFAC component intensities. A freeze/thaw experiment with water samples from a large number of UK locations conducted by Spencer et al. 2009 showed large and variable changes (decreasing and increasing) in DOM fluorescence intensity and absorbance after freezing and thawing. Likewise Peakock et al. (2015) found strong and inconsistent effects of freezing and thawing on absorbance properties of cDOM in water from bog pools, fen ditches and lakes. In a study of sample preservation on rainwater cDOM fluorescence, Santos et al. (2010) found a decrease of protein-like fluorescence intensity due to freezing. While many studies investigated the influence of different soil sample pre treatments on DOC concentrations and DOM composition (e.g. Christ and David 1994; Sun et al. 2015) only few studies focused on the influence on these properties when using different preservation methods for the extracted soil solutions. Otero et al. (2007) conducted freeze/thaw experiments on salt marsh pore water and found no changes in characteristics of synchronous fluorescence scans. The impact of sample preservation like freezing seems highly variable depending on sample and DOM characteristics. While most studies focused on samples from marine or freshwater ecosystems, there is a lack of information on sample pre-treatment effects on cDOM properties of water samples from terrestrial ecosystems, especially soil solution. Due to different sources of DOM in land and water environments (Bolan et al. 2011) and therefore different chemical characteristic, it is unlikely that insight regarding the alterations of samples during storage can be transferred from one sample type to another. To help closing this gap, we investigate in this study the influence of freezing and thawing on DOC concentration, spectral absorption and fluorescence properties for a wide range of water samples (throughfall, litter leachate and soil solution) from different terrestrial ecosystems (grasslands and forests). We tested in how far fast-freezing with liquid nitrogen might prevent concentration and partitioning effects and minimize structural changes of DOM. We hypothesized i) that sample type affects freeze/thaw effects on DOC concentrations and DOM properties, because of different physical and chemical DOM characteristics and therefore different response to changing conditions during freezing and ii) that fast-freezing with liquid nitrogen reduces these freeze/thaw effects, because it minimizes the freezing time and thus prevents partitioning effects and their physical consequences.

Sampling and sample preparation: The approach for sampling, replication, and defining the subject for the analysis needs to be clarified. The existing description is hard to follow. It might help to provide a diagram for where the samples originate and their fate, with a clear identification of what is composited and analyzed. This will clearly highlight the mixing of the grassland and forest samples.

We did not mix sample solutions from forest and grassland plots for obtaining composite samples for chemical or spectroscopic analysis. All samples were analyzed separately in the laboratory. We used replicated sampling devices per sample type

(e.g. topsoil solution or throughfall) on the individual plots (forest: W1, W2, W3, W5 and W9; grassland: G3, G5 and G39) in order to gain composite samples with sufficient volume for the experiment. We will rearrange the description of sampling and sample preparation for clarification. While the forest and grassland samples were processed separately in the laboratory, the results were analyzed in one statistical analysis. This analysis did not reveal significant differences between grassland samples and forest samples (PERMANOVA, $R^2$ = 0.05184, p = 0.2401). The freezing procedures are somewhat tedious. Is this operational? What happens if a large quantity of water is stored? Is there a potential difference given the small amounts used as test subjects in the study? The procedure for freezing samples at -18°C corresponds to the routine procedure in the above mentioned BECYCLES project. We commonly keep the sample volume that is stored frozen as small as possible because of space limitations in deep freezers. We think that increasing the volume of samples that are subjected to freezing also increases the risk of artifacts, because of increasing concentration effects due to extended freezing time. We will included a short discussion of this in the revised manuscript.

Results: The overall average change of 6% (1.6 mg L-1) seems small given the high DOC concentrations in the samples. Is the lower average a result of the composite?

This comment may be a misunderstanding of the methods applied in the study. We did not produce composite samples across different sample types, as we tried to explain with the answer of your comment to Sampling and sample preparation (above). A good point is to test the influence of the initial DOC concentrations on the changes of DOM properties due to different treatments. We found a significant correlation between initial DOC concentration of the fresh sample and the changes of DOC concentrations for the -18°C freezing treatment (Spearmans rank r = -0,447, p = 0,0194). This indicates a larger decrease of DOC concentration during freezing at -18°C for samples with higher initial DOC concentration. Additionally we run a new PERMANOVA extended with DOC concentrations (NPOC concentration in the following table) of the fresh samples as factor (see Fig.1:PERMANOVA_results) The interaction between the freezing treatment (either at -18°C or with liquid N2) and DOC concentration of the fresh sample explains a reasonable part of the variance of DOC concentrations ($R2 = 0.14$) and is highly significant. However, the fraction of the variation that is explained by the main treatment is as low as before ($R2 = 0.05$). It is important to note that the tested dependent variables of the PERMANOVA were the DOM composition variables without DOC concentration. Therefore, fast-freezing with liquid N2 still eliminates the significant effect of freezing on DOC concentrations. Altogether initial DOC concentration well explains the different strength of the effect of treatment on DOM composition. In other words, the higher the initial DOC concentration, the stronger the effect by freezing on DOM composition. But, (as before) there are more variables, which may add to the explanation. We will add the new statistic and their results to the results section.

L30: This doesn't make sense. SUVA values increase, so aromatic compounds or aromaticity increase. But, humification index decreases?

SUVA is an absorbance-based indicator, reflecting aromaticity, whereas HIX is a common indicator of humification based on low-Stokes shift fluorescence (protein-like), relative to high-Stokes shift fluorescence (humic-like). Therefore, both indicators allow different interpretations and can have opposite tendencies within a dataset. In fact, HIX is not necessarily linked to aromaticity but rather to a wavelength shift in the emission of so-called humified DOM (Fellman et al. (2010) Limnology and Oceanography 55:2452–2462). Theoretically (and in an extreme case), if a sample is only consistent of amino acids with aromatic groups (Tyrosine, Tryptophan, Phenylalanine), it could have a high SUVA but a very low HIX.

Conclusion: There needs to be some discussion of the results related to very high DOC concentrations in the sample. What are the implications for changes in the DOM character with freezing?

We will include a discussion of the results of the new statistic, showing that higher

initial DOC concentrations lead to higher losses of DOC during freezing at -18°C . This finding is consistent with results of Fellman et al. 2008 who suggested freezing as preservation method only for water samples with DOC concentrations < 5mgL-1. Our results for fast-freezing with liquid nitrogen show the opportunity of freezing as conservation method for samples with higher concentrations without altering the bulk DOC amount.

Also, is freezing with N2 practical?

Of course, in field freezing with liquid nitrogen would take some extra effort concerning material and costs and is probably only applicable for small sample volumes. The objective of our experiment was testing if the increased effort and cost of using liquid nitrogen in the field is justified by advantages regarding the minimization of freezing artifacts. We will add this consideration to the conclusion section of the revised manuscript.

Figure 1: Is cDOC an accepted convention? A label of DOC with the units usually implies a concentration. I suggest adding 'in' for Change in DOC concentration, Or DOC change

Graphs a and b in Figure 1 show the DOC concentrations of the samples before freezing them. We will change the label of the y-axis of these graphs into "DOC concentration (mg L-1)". The label of the y-axis of graphs c and d will be changed into "change in DOC conc. (mg L-1)". The label of graphs e and f will be changed into "change in DOC conc. (%)".
* * *
| | Df | SumsOfSqs | MeanSqs | F.Model | R2 | Pr(>F) |
|---|---|---|---|---|---|---|
| treatment | 2 | 32.41 | 16.2066 | 2.3584 | 0.05065 | 1e-04 *** |
| treatment:npoc.factor | 3 | 92.21 | 30.7358 | 4.4728 | 0.14407 | 1e-04 *** |
| Residuals | 75 | 515.38 | 6.8717 | | 0.80528 | |
| Total | 80 | 640.00 | | | 1.00000 | |

Signif. codes:  0 '***' 0.001 '**' 0.01 '*' 0.05 '.' 0.1 ' ' 1

**Fig. 1.** PERMANOVA_results